# Annexin A2 in Tumors of the Gastrointestinal Tract, Liver, and Pancreas

**DOI:** 10.3390/cancers16223764

**Published:** 2024-11-08

**Authors:** Konstantinos Christofidis, Alexandros Pergaris, Rodanthi Fioretzaki, Nikolaos Charalampakis, Emmanouil Ι. Kapetanakis, Nikolaos Kavantzas, Dimitrios Schizas, Stratigoula Sakellariou

**Affiliations:** 1Cytopathology Laboratory, Laiko General Hospital of Athens, 11527 Athens, Greece; cchristof@med.uoa.gr (K.C.); nkavantz@med.uoa.gr (N.K.); sakelstrat@med.uoa.gr (S.S.); 2First Department of Pathology, School of Medicine, National and Kapodistrian University of Athens, 11527 Athens, Greece; alexperg@yahoo.com; 3First Department of Surgery, National and Kapodistrian University of Athens, Laiko General Hospital of Athens, 11527 Athens, Greece; rodanthifioretzaki@gmail.com (R.F.); dschizas@med.uoa.gr (D.S.); 4Department of Medical Oncology, Metaxa Cancer Hospital of Piraeus, 18537 Piraeus, Greece; nick301178@yahoo.com; 5Department of Thoracic Surgery, National and Kapodistrian University of Athens, Attikon University Hospital, 12462 Athens, Greece

**Keywords:** annexins, ANXA2, digestive system, oncogenesis, biomarker, therapy, esophageal squamous cell carcinoma, gastric adenocarcinoma, colorectal adenocarcinoma, hepatocellular carcinoma, pancreatic ductal adenocarcinoma

## Abstract

Annexin A2 is a protein that plays a role in many important cell functions, but when its levels are not properly regulated, it can contribute to the development of cancer, particularly in the digestive system. This review looks at the current understanding of how Annexin A2 is involved in cancers of the digestive system, including its potential to help diagnose cancer, predict how a patient will respond to treatment, and even serve as a target for new therapies. Some early studies suggest that blocking this protein or stopping it from interacting with other molecules could slow down cancer growth, prevent the spread of cancer, and make cancer cells more responsive to treatment. By exploring this protein further, scientists hope to develop better tools to diagnose and fight cancer, ultimately benefiting society by improving survival rates and quality of life for patients.

## 1. Introduction

Annexin A2 (ANXA2) is a protein with diverse roles in normal cellular physiology and pathology. Its complex regulation and diverse interactions make it a captivating subject of research. The aberrant expression of ANXA2 has been implicated in the oncogenesis of multiple cancers. This review seeks to provide an overview of its role in DS tumors, including but not limited to esophageal squamous cell carcinoma (ESCC), gastric cancer (GC), colorectal cancer (CRC), hepatocellular carcinoma (HCC), cholangiocarcinoma (CCA), and pancreatic ductal adenocarcinoma (PDAC), in the hope of promoting research on the clinical applications of ANXA2.

### 1.1. Structure and Cellular Localization of Annexin A2

Annexin A2 (also called ANXA2 and Annexin II, as well as p36, calpactin I, lipocortin II, chromobindin VIII, or placental anticoagulant protein IV in older publications) [1] is a member of the highly conserved phospholipid-binding protein family of annexins that includes 160 Annexin proteins [2], with a fundamental role in the cellular biology [3] of vertebrates (ANXA), invertebrates (ANXB), fungi and protozoa (ANXC), and plants (ANXD), as well as protists (ANXE) [4]. The name Annexins comes from the ability of these proteins to aggregate (annex) membranes [5]. In humans, ANXA2 is a 36-kDa protein encoded by the *ANXA2* gene located on chromosome 15q22.2 that exhibits remarkable structural diversity, and participates in many cellular processes. It is normally expressed mainly on the cellular surface of many cell types, including epithelial cells, macrophages, and mononuclear cells [6].

ANXA2 can be found as a monomer, a heterodimer, or a heterotetramer, with each form characterized by distinct roles and localization [6,7]. The monomeric form is composed of two identical ANXA2 subunits, and is primarily localized in the nucleus and cytoplasm, as well as on early endosomes [8]. Within the nucleus, ANXA2 interacts with primer recognition proteins and participates in DNA replication and DNA polymerase alpha activity [9,10]. It also functions as a major nuclear DNA-binding protein, and influences DNA synthesis, cell cycle progression, and cell proliferation [11,12]. Notably, ANXA2 accumulation in the nucleus has been proven to protect cells from DNA damage during oxidative stress [13]. The heterodimeric form of ANXA2, consisting of one ANXA2 subunit and one S100A10 subunit, is often associated with the inner leaflet of the plasma membrane. This form participates in various signaling events, including those triggered by growth factors and cytokines. Finally, the heterotetrameric form is composed of two ANXA2 subunits and two S100A10 molecules, is stabilized by calcium ions, and can be found predominantly on the cell membrane, where it exhibits enhanced binding affinity to phospholipids.

The ANXA2 protein has three distinct domains, each having specific functions. The N-terminal domain is probably the most important one, containing the binding sites for S100A10, t-PA, and other regulatory proteins. This domain also harbors several post-translational modification sites, including three phosphorylation sites that are critical for ANXA2 biological functions [14,15]. Phosphorylation at tyrosine residue Tyr23 has been shown to induce translocation of ANXA2 to the nucleus. On the contrary, phosphorylation of Ser11 and Ser25 induces the translocation of ANXA2 away from the nucleus. p11 binding promotes its conversion to its heterotetramer form [16]. As far as the conserved core domain is concerned, it is responsible for calcium binding, causing the exposure of hydrophobic residues that consequently facilitate membrane binding. Finally, the C-terminal domain is mainly known for its interaction with F-actin [17], heparin [18], and plasminogen [19], thus its subsequent participation in cytoskeleton dynamics, extracellular matrix remodeling, and fibrinolysis.

### 1.2. Regulation of Annexin A2

ANXA2 functions are tightly regulated. Phosphorylation, nitration, acetylation, and ubiquitination are key post-translational modifications that modulate its activity, stability, and subcellular localization. As previously mentioned, phosphorylation at tyrosine and serine residues within the N-terminal domain lead to nuclear translocation, cell surface expression, and are associated with the cytoskeleton [20,21,22,23]. These modifications can be triggered by various signaling pathways, including those of tyrosine kinases and protein kinase C (PKC) [20]. Moreover, following phosphorylation and nitration of the tetramer of ANXA2, liposome aggregation is inhibited [24]. The N-terminal acetylation is needed for p11 binding to take place [25]. Abnormal ubiquitination of ANXA2 has been implicated in promoting metastasis of certain tumor types [26].

### 1.3. Physiological Functions and Pathological Implications of Annexin A2

Intracellularly, ANXA2 plays a pivotal role in multiple processes, including exocytosis [27,28], endocytosis [9,29], membrane trafficking [30], and biogenesis of polycystic transport intermediates [31]. It also contributes to cell division and proliferation by regulating DNA synthesis and interacting with key cell cycle proteins [32]. ANXA2 association with CD44, a cell surface receptor for hyaluronic acid, is essential for lipid raft formation [30] that interacts with the cellular cytoskeleton [33] and signal transduction, influencing cellular responses to external stimuli. Furthermore, intracellular ANXA2 has been shown to participate in the prevention of radiation-induced apoptosis [13,34], and in the regulation of inflammatory responses [35,36,37,38]. On cells going through apoptosis, ANXA2 acts as a C1q ligand [39]. Moreover, the degradation of ANXA2 can lead to cell apoptosis via certain molecular pathways, where p53 plays a pivotal role [40].

Extracellularly, ANXA2 functions are equally diverse and important. It participates in phagocytosis [27,39], fibrinolysis, anticoagulation [41,42], and angiogenesis [42,43]. The ANXA2/S100A10 heterotetramer interaction with tenascin-C, plasminogen, and tPA, is important for the conversion of plasminogen to plasmin and the fibrinolytic homeostasis [41,44,45,46,47]. Moreover, ANXA2 seems to play a role in the immune system, influencing inflammatory cytokines secretion [48,49,50].

Annexins play such a vital role in many human diseases that the term “annexinopathies” has been coined. In most types of cancer, ANXA2 develops its pro-oncogenic role by some of the mechanisms shown in Table 1 [1,3,4,51,52,53,54,55,56,57,58]:

Given its multifaceted roles in cancer progression (Figure 1), ANXA2 has emerged as a promising target for therapeutic intervention. Various therapeutic techniques have been exploited with the hope of inhibiting the carcinogenic effects of ANXA2 [3,58,59,60]. In addition to that, ANXA2 is continuously being tested as a potential diagnostic and prognostic biomarker for many cancers, something that could significantly help in patients’ management [1,3,58,59,60].

In a nutshell, ANXA2 is a protein playing vital roles in cellular physiology and pathology. Its complex regulation, diverse interactions and involvement in numerous cellular processes make it a captivating subject of research. The latter may lead to novel scientific insights and clinical applications capable of improving human health. In that direction, the current study will summarize the role of ANXA2 in DS tumors.

## 2. Methods

PubMed and Google Scholar were last searched on 9 June 2024, with the following search strings: (Annexin A2 OR ANXA2) AND (functions OR role) AND (gastrointestinal track OR digestive system OR esophagus OR stomach OR colon OR rectum OR liver OR pancreas OR cancer OR pathology). Only publications available in English were screened, with no restriction as to the time or design of the studies.

Two independent authors (KC and SS) reviewed all of the titles/abstracts, and selected studies meeting the inclusion criteria. Specifically, original studies and reviews in English language concerning the biological functions of ANXA2 in GI tract, liver, and pancreas in humans were eligible for inclusion. Letters, comments, editorials, and case reports, as well as abstracts or conference abstracts without the full text available, and studies written in languages other than English, were excluded from this review.

The following data were extracted by KC and SS independently: (I) publication details (authors, year of publication, references); (II) basic characteristics (samples, controls, materials, methods); (III) results; and (IV) specific signaling pathways.

## 3. The Role of Annexin A2 in DS Tumors

### 3.1. Annexin A2 in Tumors of the Esophagus

There have been only a few studies of ANXA2 in esophageal tumors, which present a complex picture with seemingly contradictory findings regarding its role in esophageal squamous cell carcinoma (ESCC).

Contrary to most organs of the DS, in esophageal cancers ANXA2 seems to be downregulated in most cases. Qi YJ et al. conducted a study in a region of China with a high incidence of ESCC, and found ANXA2 to be markedly downregulated in ESCC tissues compared to adjacent normal ones. This was particularly pronounced in advanced stage tumors. These findings align with the hypothesized tumor suppressor role of ANXA2 and, furthermore, demonstrate the possible use of ANXA2 for early ESCC detection [59]. Feng JG et al. further corroborated this potential of ANXA2, alongside Cdc42, as ESCC prognostic biomarkers, in a study that found that downregulation of ANXA2 and upregulated Cdc42 expression significantly correlated with the presence of lymph node metastasis and poor tumor differentiation, both hallmarks of aggressive tumor behavior [60]. Li X et al. investigated ANXA2 expression in ESCC of the Kazakh population, and they similarly found a marked down-regulation of ANXA2 in ESCC tissues. Importantly, they observed that the degree of ANXA2 downregulation correlated with increased tumor invasiveness and loss of differentiation. To delve even deeper, the authors performed in vitro experiments, and demonstrated that ANXA2 overexpression in ESCC cells suppresses their invasion and migration potential, as well as their proliferation, arresting the cell cycle at the G2 phase, although without causing apoptosis. Based on these findings, it seems probable that ANXA2 could suppress ESCC development [61].

In contrast to the aforementioned studies, ANXA2 has been proved to play a pro-oncogenic role in ESCC by other researchers. Notably, Ma RL et al. studied human ESCC tissues and cell lines, and showed that ANXA2 and SOD2 are potential target genes of HOXA13. Furthermore, the co-expression of HOXA13/ANXA2/SOD2 is an independent prognostic factor linked to poor prognosis in ESCC [62]. Cao HH et al., in a study with human ESCC tissues, proposed a prognostic model that could predict patient survival and tumor recurrence. In this model, the expression of ANXA2, kindlin-2, and myosin-9 significantly correlated with a poor overall and disease-free patient survival [63]. In addition, Ma S et al. focused on the phosphorylated form of ANXA2 at tyrosine 23 in human ESCC tissues, cell lines, and xenograft tumors. They discovered that its overexpression promoted ESCC cell migration, invasion, and metastasis. This suggests that p-ANXA2 (Tyr23) may promote ESCC progression. The authors also demonstrated that this phosphorylated form of ANXA2 activates the MYC-HIF-1α-VEGF signaling axis, inhibition of which hampers the growth of ESCC xenograft tumors. These results point to the potential therapeutic benefit of targeting ANXA2 or other molecules in this pathway [64].

To date, there is no accepted explanation for these seemingly contradictory findings. Overall, the role of ANXA2 in ESCC largely remains terra incognita. However, the current research highlights its value as a diagnostic and prognostic biomarker [59,60,61,62,63,64], as well as a potential therapeutic tool [64]. ANXA2 thus warrants additional research in order to unravel and understand the intricate mechanisms governing its contribution to ESCC development. The following table summarizes existing research on the role of ANXA2 in esophageal tumors (Table 2).

### 3.2. Annexin A2 in Tumors of the Stomach

The research of the role of ANXA2 in gastric tumors has been more extensive, with several studies reporting its up-regulation in GC tissues compared to normal tissues, which they associated with various clinicopathological features of GC and a poor prognosis.

Studying human GC tissues and normal gastric epithelium adjacent to the tumors, Zhang Q. et al. demonstrated that ANXA2 and S100A6 were both up-regulated in GC. The expression of the former correlated with tumor depth of invasion, lymph node and distant metastasis, TNM stage, high S100A5 expression, and poor prognosis, suggesting that ANXA2 and S100A6 may serve as prognostic indicators in GC patients [65]. Han Y. et al. proved that both ANXA2 mRNA and protein concentrations were higher in gastric adenocarcinoma (GAC) tissues. They also demonstrated a link between this upregulation and the tumor’s stage and metastatic potential, as well as a negative correlation between the expression of ANXA2 and E-cadherin, further proving that ANXA2 may contribute to metastasis in GAC, and that it can represent a potential target for treatment [66]. In addition to the above, Han F. et al. showed that inhibiting ANXA2 expression decreased GC cell migration and secretion of matrix metalloproteinases [67]. In a study by Leal MF et al. in GAC tissues and cell lines, ANXA2 was found to be up-regulated, while GAL3 was down-regulated, leading to increased invasive and metastatic potential of GC [68].

In another study by Zhang ZD et al., ANXA2 was researched in human GAC tissues and cell lines and was found to be significantly up-regulated in GAC compared to normal tissues, negatively correlating with the differentiation level of GAC. Its expression was also higher in cisplatin (DDP)-resistant cells compared to parental cells. Moreover, silencing ANXA2 decreased the phosphorylation of p38MAPK and AKT, key components of signaling pathways involved in drug resistance, and thus increased drug sensitivity of DDP-resistant cells, indicating that targeting ANXA2 could be a potential strategy to overcome multidrug resistance [69]. Xie R et al. utilized The Cancer Genome Atlas (TCGA) database, GC tissue microarrays, and cell lines, and demonstrated that ANXA2 was upregulated in GC tissues at both the transcriptional and translational levels, promoting GC cell migration and proliferation. When they silenced ANXA2, the proliferation, invasion, and migration potential of GC cells were diminished, and their apoptotic rate increased, further highlighting ANXA2 value as a therapeutic target [70]. Finally, Mao et al. explored the role of the EphA2-YES1-ANXA2 pathway in GC using GC cell lines and also mouse models. They found that this pathway drives GC invasion and metastasis, and targeting it could be an effective treatment strategy for GC [71].

The existing literature is not limited to studies of GC tissues and cell lines. Patient serum samples were analyzed by Tas et al., who found their ANXA2 levels to be higher than in healthy controls. Interestingly, patients whowere unresponsive to chemotherapy had higher levels compared to those that responded to chemotherapy, suggesting that ANXA2 may serve as a predictive biomarker for patients’ response to chemotherapy [72].

In summary, despite the limited research existing to date, ANXA2 seems to be a very promising molecule that, if researched more extensively, could find multiple clinical applications, as a diagnostic and prognostic biological marker [65,67,68,69,72] or as part of therapeutic interventions [66,67,69,70,71]. Table 3 presents the existing research on the role of ANXA2 in tumors of the stomach.

### 3.3. Annexin A2 in Tumors of the Colorectum

Annexin A2 has been extensively studied in the colorectum and has emerged as a significant factor in the oncogenesis and progression of human colorectal neoplasms, as well as a potentially valuable diagnostic and therapeutic tool.

Many studies found ANXA2 to be over expressed in CRC tissues compared to normal tissues [73,74,75,76]. In addition to that, Emoto et al. and Yang et al. showed that high ANXA2 expression correlated with tumor size, depth of invasion, and pTNM stage, insinuating its value as a prognostic biomarker in CRC [73,75]. Emoto et al. also demonstrated that the expression of ANXA2 was correlated with that of tenascin-C, and that their co-overexpression was an independent poor prognostic factor in CRC [73]. Mousa et al. additionally examined the expression of ANXA2 in tissues from colorectal adenomas. They found that ANXA2 expression was increased in adenomas with high-grade dysplasia, especially of the tubulovillous type, suggesting that it may be involved in the early stages of CRC oncogenesis. The expression of ANXA2 in adenomas, even in the high-grade dysplastic ones, was still lower than in CRC [76].

Several studies have focused on the interactions between ANXA2 and other molecules. Singh et al. and Sarkar et al. examined the relationship between ANXA2 and gastrin-like peptides in colon cancer (CC) cell lines. The former found that ANXA2 binds to these peptides and mediates their effects [77], while the latter showed that ANXA2 facilitates the uptake of progastrin into cells via the clathrin-mediated endocytic pathway, resulting in activation of MAP Kinases that promote tumor cell growth and survival. In the absence of clathrin and/or cell surface-associated ANXA2, progastrin failed to activate the p38MAPK/ERKs [78]. In a study of CC tissues and cell lines by Rocha et al., it was demonstrated that overexpression of ANXA2 plays a pivotal role in CRC invasiveness through the activation of the Src/ANXA2/STAT3 pathway. Upon TGF-ß treatment, ANXA2 and E-cadherin co-localize at cellular junctions and induce EMT after being internalized together [79]. Zhao et al. showed that TAGLN2 activates ANXA2 and STAT3 signaling, facilitating EMT, CRC cell proliferation, migration and invasion [80]. Hong et al. and Hosseini et al. both found the long noncoding RNA LINC00460 to regulate ANXA2 expression. Hong et al. proved that silencing LINC00460 leads to decreased ANXA2 expression via miR-433-3p, and inhibits the proliferation, invasion, and tumorigenesis of CC cell lines. Additionally, the LINC00460/miR-433-3p/ANXA2 axis was shown to regulate EMT in CC [81]. Hosseini et al. found that both ANXA2 and LINC00460 were overexpressed in tissue from colorectal polyps, particularly in high-risk polyps, and that their expression was associated with poor clinical outcomes [82].

A study by Guzmán-Aránguez et al. in human colorectal adenocarcinoma (CAC) cell lines demonstrated that the dedifferentiation of CAC cells is associated with the up-regulation of ANXA2 and its subcellular relocation [83], while Tristante et al. found that a high number of ANXA2-positively labeled (membranous pattern of expression) in the tumor invasive front was associated with high invasiveness and lymph node metastasis of CRC [84]. Studying human CC cell lines, Xing et al. showed that ANXA2 remodels the cytoskeleton of CC cells, augments the expression and polymerization of F-actin and β-tubulin and counteracts cell contact inhibition, leading to increased cell motility and invasiveness [85]. This is in agreement with He et al., who proved that ANXA2 enhances the malignancy of cancer cells by remodeling the cytoskeleton and promoting cell motility. The latter also found that ANXA2 regulates various signaling pathways involved in cell proliferation and survival [86]. Xiao et al. demonstrated that specific phosphorylation sites in the ANXA2 protein are important for its function, and that their mutation can inhibit the malignant behavior of CRC cells. Notably, both in vitro and in vivo, the malignant potential of CRC was limited by ANXA2 silencing, strengthening the belief in the therapeutic potential of targeting ANXA2 [87].

Finally, serum ANXA2 was investigated by Hu et al. and Gurluler et al. as to its potential use as a diagnostic biomarker for CRC. Interestingly enough, both studies found that, compared to healthy controls, ANXA2 serum levels were significantly lower in CRC patients, while the opposite was true for CA19.9 and CEA according to Hu et al. [88]. Gurluler et al. additionally demonstrated that ANXA2 serum concentration decreased even further with increasing tumor size, TNM stage, tumor invasion, lymph node metastasis, and distant metastasis [89].

The colorectum is the most extensively researched part of the gastrointestinal system regarding the role of ANXA2 in tumorigenesis. Multiple studies have repeatedly proven the value of ANXA2 as a biomarker [73,74,75,76,82,84,85,88,89], as well as a supplement to the current treatments [78,81,85,87]. Table 4 depicts the studies having researched the role of ANXA2 in colorectal tumors.

The oncogenic mechanisms activated by ANXA2 in tumors of the main organs of the gastrointestinal tube are presented in Figure 2.

### 3.4. Annexin A2 in Tumors of the Liver

ANXA2 has also been studied in the context of liver cancers, particularly in HCC and, less so, in CCA, with research highlighting its complex oncogenic role and its potential use as a biomarker or a therapeutic tool.

The overexpression of ANXA2 in HCC tissues compared to cirrhotic liver tissues was first reported by Yu et al. The authors found ANXA2 to be overexpressed in the sinusoidal endothelium and in malignant hepatocytes, and suggested that ANXA2 may contribute to HCC angiogenesis [90]. Shortly after that, Mohammad et al. also confirmed the overexpression of ANXA2 in HCC compared to non-tumorous liver tissues, while additionally demonstrating that, in HCC, ANXA2 is tyrosine-phosphorylated and overexpressed at both the transcriptional and translational levels [91]. Longerich et al., after studying tissues from human hepatocellular nodules of various etiologies, suggested that adding ANXA2 (sinusoidal pattern of expression) to the currently used selection of markers for diagnosing well-differentiated HCC boosts liver biopsies’ diagnostic accuracy. Specifically, the combinations ANXA2-GPC3 and ANXA2-GS were shown to have the highest diagnostic value [92]. Zhang et al. and Dong et al., after respectively examining HCC cell lines and tissues, as well as mouse models, showed HCC cells with high metastatic and invasive potential had higher levels of ANXA2 expression, and that silencing of the latter diminished their invasion, migration, and tumorigenic potential [93,94]. In another study, Zhang et al. found that ANXA2 expression was frequently up-regulated in the tissues, as well as in the serum of HCC patients and correlated with intrahepatic metastasis, portal vein thrombosis, and higher TNM stages. The authors suggested that ANXA2 acted as an independent adverse prognostic factor in HCC [95]. Other studies have also confirmed the overexpression of ANXA2 in HCC compared to non-tumorous liver tissues, as well as in the serum of HCC patients compared to normal individuals [96,97,98]. Sun et al. suggested that serum ANXA2 may even be an independent and discriminative serological biomarker of HBV-related HCC, especially in early-stage cases with normal serum AFP [97]. Shaker et al. also suggested that serum ANXA2 may serve as a biomarker for the early detection of HCC [98].

The molecular mechanisms by which ANXA2 contributes to HCC progression have also been investigated to some degree. Kittaka et al. showed that ANXA2 was one of the key members linked to the integrin and Akt/NF-kB signaling pathways in HCC. They also proved the colocalization of ANXA2 and S100A10 in human HCC tissues [99]. Zhao et al. research on human HCC cell lines proved that the interaction of HAb18G/CD147 with ANXA2 is involved in the mesenchymal and amoeboid movement of HCC cells, influencing their metastatic potential and invasion ability. They also offered clues to the involvement of the RhoA/ROCK, Rac1/WAVE2 and integrin-FAK-PI3K pathways in the oncogenesis of HCC [100]. In the same vein, Zhang et al. and Cui et al. demonstrated that CD147, via ANXA2 and theDOCK3-β-catenin-WAVE2 signaling axis was linked to HCC metastasis [95]. In the study of Cui et al., the interaction of CD147 with ANXA2 decreased the phosphorylation of the latter. Notably, p-Annexin A2 increases DOCK3 expression which inhibits β-catenin signaling and the expression of WAVE2 [101]. Finally, Bai et al. found that UBAP2 is significantly downregulated in HCC cells and that its overexpression promotes ubiquitination and degradation of ANXA2, thus impairing the progression of HCC. UBAP2 appears as a potential novel prognostic marker as well as a therapeutic target for HCC [102].

The role of ANXA2 in CCA has not been extensively investigated. Yonglitthipagon et al. demonstrated that ANXA2 is overexpressed in CCA tissues linked with Opisthorchis Viverrini infection, and that its high expression was associated with metastasis, lymphatic and perineural invasion, poor prognosis, and shorter survival time, but not with the histological subtype or the tumor size of CCA. The authors proposed that ANXA2 may serve as a prognostic biomarker in O.viverrini-associated CCA [103]. In Table 5, the existing studies on the role of ANXA2 in tumors of the liver are presented.

Similarly to the organs of the gastrointestinal tube, ANXA2 seems to play a vital role in liver carcinogenesis and can thus be exploited to widen our therapeutic [93,94,100,101,102] and diagnostic modalities, as well as improve the assessment of patients’ prognosis [91,92,93,94,95,96,97,98,102].

### 3.5. Annexin A2 in Tumors of the Pancreas

ANXA2 also seems to participate in the development and progression of pancreatic cancer (PC). Numerous studies have investigated ANXA2 in PDAC underlying its primordial role.

ANXA2 seems to be overexpressed in PDAC tissues compared to the normal pancreas in many studies [104,105,106,107,108], suggesting that ANXA2 may be a potential biomarker for PC. Huang et al. also found ANXA2 to be associated with the pathological grade and the microvessel density of PDAC, the latter being an independent risk factor of overall and disease-free survival [106].

Díaz et al. and Nedjadi et al. studied the interaction of ANXA2 with other molecules in human PC cell lines. The first showed that PC cell invasion is aided by the binding of t-PA to ANXA2, and the subsequent increase in plasmin production [109], while the second found a positive correlation between high levels of cytoplasmic S100A6 and the localization of ANXA2 to the cell membrane, where it promotes PC cell motility [110]. Esposito et al. and Yoneura et al. demonstrated that the interaction between ANXA2 and Tenascin C may play a role in pancreatic carcinogenesis. More specifically, Esposito et al. demonstrated that moving from low-grade pancreatic intraepithelial neoplasia (PanIN) to PC, the cellular expression of Tenascin C and cell-surface ANXA2 progressively increases, without an accompanying increase in their serum concentration [111]. Yoneura et al. proved that ANXA2 is overexpressed in murine PC cells, and that its interaction with Tenascin C promotes EMT, invasion, and metastasis. High expression of ANXA2 and stromal Tenascin C was shown to positively correlate with distant metastasis, hematogenous and peritoneal recurrence, poor prognosis, as well as anoikic resistance in PDAC [108].

On a different note, Takano et al. and Jung et al. suggested that ANXA2 may be involved in gemcitabine resistance in PC cells. Takano et al. demonstrated that ANXA2 was up-regulated in the gemcitabine-resistant PC cell lines. In postoperative patients, after gemcitabine adjuvant chemotherapy, ANXA2 overexpression, as well as a high degree of ANXA2 staining in the cancer cells, correlated with a rapid recurrence. Notably, silencing of ANXA2 increased gemcitabine cytotoxic effects [112]. Jung et al. additionally found that ANXA2 interacts with the p50 subunit of NF-κB, increasing its transcriptional activity and up-regulating anti-apoptotic genes, like IL-6. They also showed that following ANXA2knockdown, the activity of the NF-κB pathway and the viability PDAC cells are hindered [113]. ANXA2 could thus be considered as a biomarker for gemcitabine drug resistance and as a candidate target in the treatment of PDAC.

Studies on animal models were also carried out, in order to further elucidate the in vivo importance of ANXA2 [107,114,115,116,117]. Zheng et al. demonstrated that cell surface ANXA2 increases with PDAC development and progression, and that its phosphorylation at Tyr23 promotes its localization in the cell surface, which is required for the TGFβ-induced, Rho-mediated EMT of PDAC cells that increases their invasion capacity. Moreover, the expression and tyrosine phosphorylation of ANXA2 were shown to be necessary for PDAC metastases in vivo, and that the inhibition of ANXA2 suppresses in vitro invasion and in vivo metastases of PDAC, prolonging survival in a mouse model of PDAC. Interestingly, it was also shown that serum from PDAC patients receiving an allogeneic GM-CSF secreting tumor vaccine (GVAX) inhibited in vitro invasion of PDAC cells. All of these point to the potential use of ANXA2 as a PDAC biomarker and an antigenic target for the development of both therapeutic antibodies and T-cell immunotherapy [114,115]. Further evidence of the potential value of ANXA2 as a therapeutic target for PC was provided by Foley et al. and Keklikoglou et al. The former found that ANXA2 is essential for PDAC metastasis in a transgenic mouse model of PDAC, and that the gene expression, as well as the secretion of semaphorin 3D, is increased in PDAC, inducing its invasion capacity. Knockdown of semaphorin 3D decreased invasion and metastatic capacity of PDAC cells, and prolonged survival of PDAC-bearing mice. This research group also studied the serum of PDAC patients, and found antibodies against ANXA2 in some of them after surgical resection of PDAC. These antibodies seem to be associated with longer recurrence-free survival [116]. Keklikoglou et al. proved that KRAS and ANXA2 are up-regulated in PDAC cell lines, and their expression inversely correlates with miR-206 expression, which targets multiple oncogenic routes involving the KRAS-induced NF-κB signaling, ANXA2, and VEGFC. By using in vitro and in vivo approaches, they demonstrated that the re-expression of miR-206 in PDAC cells can inhibit tumor angiogenesis, delaying tumor growth and progression. miR-206 may thus be an attractive candidate for miRNA-based anticancer therapies [107]. Murphy et al. found that the disease-free and overall survival were reduced in case of high ANXA2 expression in tumor stroma. Moreover, B6 mice injected with KPC cells compared to those injected with KPCA cells showed an importantly shorter median survival, whereas in the ANXA2 KO mice, no difference in survival was noted [117]. These results further support the use of ANXA2 as a prognostic biomarker in patients with PDAC.

PC remains a challenge for modern medicine, with no adequately effective diagnostic procedures and therapeutic options available to date. The research mentioned above hints to the potential use of ANXA2 as a biomarker for the diagnosis and risk stratification of PC patients [104,105,106,108,111,112,113,114,115,117], as well as a supplement to their treatment [112,113,114,115,116]. Table 6 presents a summary of the current research on the role of ANXA2 in tumors of the pancreas.

Figure 3 illustrates the plethora of oncogenic procedures upregulated by ANXA2 in liver and pancreatic tumors.

## 4. Therapeutic Implications of Annexin A2

As already noted, there is significant evidence that ANXA2 can be utilized in the therapy of many DS tumors. In the esophagus, despite the limited existing research, ANXA2 has been shown to activate the MYC-HIF-1A-VEGF signaling in ESCC cells, a pathway whose inhibition suppresses the growth of ESCC xenograft tumors in mice [64]. In GC, the metastatic potential of GC cells can be diminished by targeting ANXA2. Research has found the expression of ANXA2 to be negatively corelated with that of e-cadherin [66], and positively correlated with the secretion of matrix metalloproteinases of GC cells [67], all of which contribute to the meta-static potential of GC cells. The proliferation, invasion, and migration capacity of GC cells were all decreased upon ANXA2 silencing, while their apoptosis increased [70]. Part of the prooncogenic effects of ANXA2 are carried out via the EphA2–YES1–ANXA2 axis, whose targeting might be a valid therapeutic strategy [71]. The silencing of ANXA2 has also been found to increase the drug sensitivity of GC cells to DDP by the decrease in phosphorylation of P38MAPK and AKT, the downregulation of P-gp, MRP1, and Bcl-2, as well as the upregulation of Bax in tumor cells [69]. In CRC, ANXA2 has been more extensively studied, and thus there is more evidence on its potential clinical utility as a therapeutic tool. ANXA2 increases the invasive and metastatic potential of CRC cells by EMT via the LINC00460/miR-433-3p/ANXA2 Axis [81], by increasing the expression and polymerization of F-actin and β-tubulin and suppressing cell contact inhibition [85], as well as by affecting specific motility-associated micro-structures of cancer cells [87]. Consequently, the targeting of ANXA2 can directly hamper all these, decreasing the oncogenic potential of CRC cells. The link of ANXA2 to progastrin has also been researched, with the former proving to be necessary for the clathrin-mediated endocytosis of progastrin, which is a prerequisite for the activation of p38MAPK/ERKs and the tumorigenic effects of progastrin to take place. Thus, ANXA2 targeting can indirectly abolish the tumorigenic effects of progastrin [78]. When it comes to liver oncology, silencing of ANXA2 arrested the cell cycle in vitro, inhibited tumor growth in vivo [94], and suppressed the invasion, migration, and tumorigenic potential of HCC cells [93]. HAb18G/CD147 and WAVE2, as well as the DOCK3-β-catenin-WAVE2 and the Integrin-FAK-PI3K/PIP3 signaling pathways, are all closely linked to ANXA2 and the movement and metastasis of HCC cells, and could thus be exploited therapeutically [100,101]. UBAP2 has been shown to promote the ubiquitination and degradation of ANXA2, hampering the progression of HCC [102]. Finally, In PC, overexpression of ANXA2 is hypothesized to contribute to the gemcitabine resistance of PC cells. Inhibition of ANXA2 affects the transcriptional activity of NF-κB and increases the cytotoxic effect of gemcitabine on gemcitabine-resistant PC cells [112,113]. Research has shown that ANXA2 is required for PDAC metastases in vivo, and that its inhibition by antibodies from post-vaccination patient sera suppresses the in vitro invasion and in vivo metastases of PDAC [114,115]. The value of anti-ANXA2 antibodies in the sera of PDAC patients ANXA2 was also researched in another study that showed that ANXA2 associated with semaphorin 3D and PlxnD1, increasing the invasive capacity of PDAC cells. ANXA2 and semaphorin 3D can thus be targeted as part of an adjuvant treatment for PDAC [116]. Despite all these promising findings, ANXA2 has not been studied in large patient series that could pave the way for it to be used not only as a diagnostic and prognostic biomarker, but also as part of therapeutic schemes.

## 5. Conclusions

Given the above literature review, the role of ANXA2 in tumors of the DS seems primordial; albeit complex and diverse, across different types and stages of DS neoplasms.

In esophageal cancer, the role of ANXA2 is not yet well studied, with research coming up with seemingly contradictory findings. While some studies suggest that ANXA2 down-regulation may contribute to ESCC progression, others have linked its overexpression to a poor clinical outcome. Its phosphorylated form, p-ANXA2 (Tyr23), may promote tumor growth and metastasis through the MYC-HIF-1α-VEGF signaling axis. These controversial findings regarding ESCC have not, to this day, found an accepted explanation. In gastric, colorectal, liver, and pancreatic cancer, most studies have found ANXA2 to be overexpressed, promoting oncogenesis by interacting with other molecules and signaling pathways; in GC, namely the p38MAPK and AKT pathways; in CRC, gastrin-like peptides, tenascin-C, LINC00460, as well as the Src/ANXA2/STAT3 pathway; in HCC integrins, CD147, UBAP2 and the RhoA/ROCK, Rac1/WAVE2 and integrin-FAK-PI3K signaling pathways; and in PC, tenascin-C, semaphorin 3D, as well as the KRAS-induced NF-κB signaling, and the TGFβ-induced, Rho-mediated EMT pathway. ANXA2 seems to also play an important role in colorectal adenomas with high-grade dysplasia, as well as in CCA.

In summary, ANXA2 has earned its place as one of the protagonists in the onco-genesis of many DS tumors. Since ANXA2 has been shown to be a vital part of specific molecular pathways characteristic of tumors’ molecular signatures, further could eventually improve the diagnosis, prognosis, and treatment of many neoplasms of the DS.

## Figures and Tables

**Figure 1 cancers-16-03764-f001:**
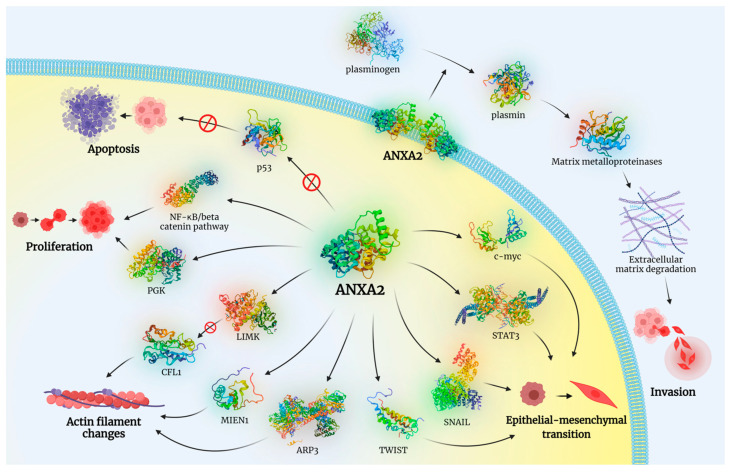
ANXA2 is implicated in a plethora of molecular pathways leading to tumorigenesis. Among others, ANXA2-induced c-myc, STAT3, SNAIL, and TWIST activation promotes the epithelial–mesenchymal transition of tumor cells. Activation of ARP3, MIEN1, and LIMK (that, in turn, inactivates CFL1) induces changes in actin filaments and the cytoskeleton, promoting cell motility. Activation of PGK and the NF-κB/beta catenin pathway results in increased cell proliferation and inhibition of p53 protein prevents apoptosis. In addition, ANXA2 catalyzes the conversion of plasminogen to plasmin which, in turn, activates matrix metalloproteinases that promote the degradation of extracellular matrix, allowing tumor cells to invade deeper in tissues. Created with BioRender.com. Abbreviations: STAT3: signal transducer and activator of transcription 3; SNAIL: zinc finger protein SNAI1; LIMK: LIM domain kinase; CFL1: Cofilin 1; ARP3: actin-related protein 3; MIEN1: Migration and Invasion Enhancer 1; PGK: 3-phosphoglycerate kinase, NF-κB: nuclear factor kappa-light-chain-enhancer of activated B cells.

**Figure 2 cancers-16-03764-f002:**
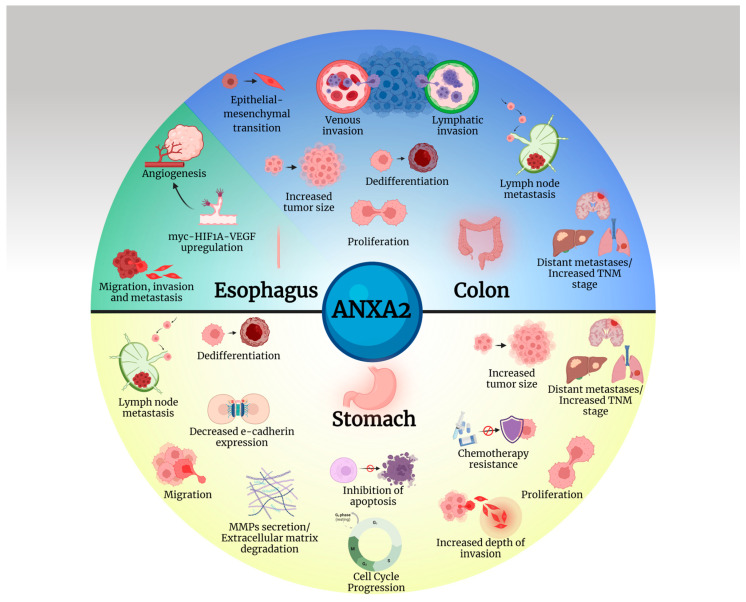
A multitude of tumorigenic procedures are promoted by ANXA2 in the organs of the gastrointestinal tract. Created with BioRender.com.

**Figure 3 cancers-16-03764-f003:**
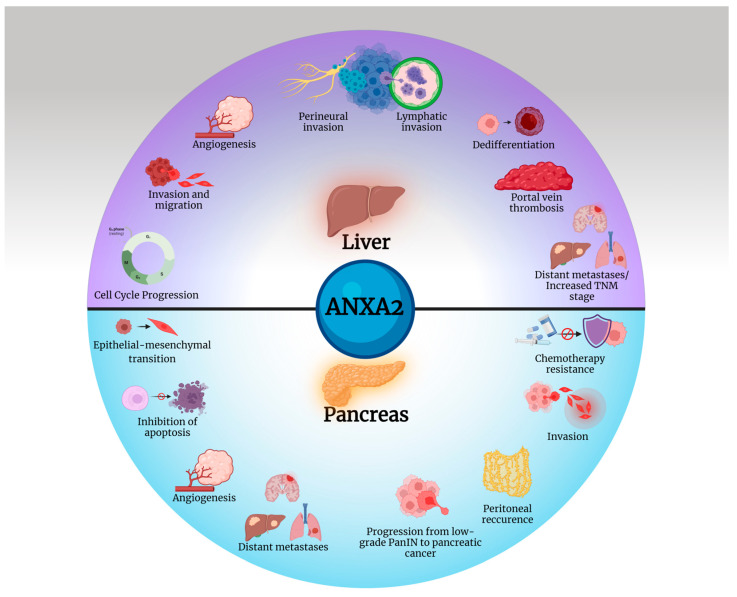
A multitude of tumorigenic procedures are promoted by ANXA2 in the liver and pancreas. Created with BioRender.com.

**Table 1 cancers-16-03764-t001:** The pro-oncogenic role of ANXA2.

Promotion of epithelial–mesenchymal transition (EMT) of cancer cells
Degradation of the extracellular matrix (ECM) by the conversion of plasminogen to plasmin
Promotion of tumor angiogenesis
Changes in the dynamics of the cytoskeleton and tumor cell motility
Regulation of cell cycle and apoptosis, with enhancement of cancer cell proliferation
Effects on tumor immunology
Development of therapeutic resistance to chemotherapy, radiation therapy, and targeted therapy

**Table 2 cancers-16-03764-t002:** The role of ANXA2 in esophageal tumors.

Ref.	Samples, Controls	Materials, Methods	Results
[59]	Human ESCC tissues Normal tissues adjacent to the tumors	Pathological analysis Immunohistochemistry RNA isolation and RT-PCR	ANXA2 was expressed in 90% of normal esophageal squamous epithelium and decreased with ESCC progression ANXA2 protein was increased in well differentiated ESCC and decreased with loss of differentiation of ESCC Up-regulation and down-regulation of ANXA2 may, respectively, correlate with regression and progression of ESCC ANXA2 may be used as a biomarker for detection and early diagnosis of ESCC, as well as for screening of high-risk patients
[60]	Human ESCC tissues Normal esophageal epithelium adjacent to the tumors	Pathological analysis Immunohistochemistry RNA isolation and qRT-PCR Western blot	ANXA2 and Cdc42 expressions were, respectively, down-regulated and up-regulated in ESCC compared to matched control ANXA2 and Cdc42 expressions significantly correlated with lymph node metastasis and pathological differentiation Aberrant ANXA2 and Cdc42 expression played a role in carcinogenesis, differentiation, and metastasis of ESCC ANXA2 and Cdc42 may serve as prognostic biomarkers of ESCC
[61]	Human ESCC tissues Normal esophageal epithelium adjacent to the tumors Human ESCC cell line (Eca109)	Pathological analysis Immunohistochemistry RNA isolation and qRT-PCR Western blot Protein Extraction and 2-Dimensional Electrophoresis (2DE) MALDI–TOF–MS analysis Cell Proliferation, Migration, and Invasion Assays Cell Cycle and Apoptosis Analysis	ANXA2 was significantly downregulated in ESCC and decreased with increasing depth of invasion and loss of differentiation Overexpression of ANXA2 can suppress proliferation, invasion, and migration Up-regulating ANXA2 caused cell cycle arrest at the G2 phase, but no apoptosis
[62]	Human ESCC tissues Normal esophageal epithelium adjacent to the tumors Human ESCC cell lines	Pathological analysis Immunohistochemistry RNA isolation and qRT-PCR Western blot	ANXA2 and SOD2 are potential target genes of HOXA13 TNM stage and co-expression of HOXA13/ANXA2/SOD2 are independent predictors of overall survival of ESCC patients
[63]	Human ESCC tissues	Pathological analysis Immunohistochemistry Tissue Microarrays	Expression of ANXA2, kindlin-2, and myosin-9 is highly predictive of poor overall survival and disease-free survival in ESCC
[64]	Human ESCC tissues Normal esophageal epithelium adjacent to the tumors Human ESCC cell lines (KYSE30, KYSE70, KYSE150, KYSE180, KYSE410, KYSE450 and KYSE510)	Pathological analysis Immunofluorescence RNA isolation and qRT-PCR Western blot Transfection and lentiviral transduction Wound-healing, Luciferase reporter, Cell matrigel migration and invasion, Chromatin immunoprecipitation Assays Co-immunoprecipitation NOD/SCID and BALB/c mice	Overexpression of the phosphorylated ANXA2 (Tyr23) promotes ESCC migration, invasion, and metastasis ANXA2 activates MYC-HIF-1A-VEGF signaling in ESCC cells Inhibition of SRC-ANXA2-MYC-HIF1A-VEGF signaling suppresses the growth of ESCC xenograft tumors, and may offer a potential therapeutic strategy for ESCC

**Table 3 cancers-16-03764-t003:** The role of ANXA2 in gastric tumors.

Ref.	Samples, Controls	Materials, Methods	Results
[65]	Human GC tissues Normal gastric epithelium adjacent to the tumors	Pathological analysis Immunohistochemistry RNA isolation and qRT-PCR Tissue microarray	ANXA2 and S100A6 were upregulated in GC Expression of ANXA2 in GC was associated with depth of invasion, lymph node, and distant metastasis, TNM stage, high S100A6 expression, and poor prognosis ANXA2 and S100A6 may be used as prognostic and survival indicators
[69]	Human GC tissues Human GC cell lines (MKN28, SGC7901 and BGC823) Human gastric mucosa epithelial cell line (GES-1) Human GC Cisplatin (DDP)-resistant cell line (SGC 7901/DDP)	Pathological analysis Immunohistochemistry RNA isolation and qRT-PCR siRNA Treatment Sulfo-rhodamine B (SRB) assay Western blot	ANXA2 expression was higher in GC than in normal tissues and negatively correlated with the differentiation level of GC ANXA2 expression level was higher in SGC7901/DDP cells than in parent SGC7901 cells Silencing of ANXA2 decreased the phosphorylation of P38MAPK and AKT, downregulated the expression of P-gp, MRP1, and Bcl-2, and upregulated Bax in SGC7901/DDP cells ANXA2 siRNAs, as well as p38MAPK and AKT inhibitor increased the drug sensitivity of SGC701/DDP cells Resistance of GAC cells to DDP could be diminished by silencing ANXA2
[72]	Serum of patients with GC that had not received chemotherapy or chemo-radiation in the last 6 months Serum from age and sex matched healthy controls	ELISA	ANXA2 serum levels of GC patients were higher compared to healthy controls Chemotherapy-unresponsive patients had higher serum ANXA2 levels compared to chemotherapy-responsive ones ANXA2 may be a good diagnostic and predictive biomarker for response to chemotherapy in patients with GC
[66]	Human GAC tissues Normal gastric epithelium adjacent to the tumors	Pathological analysis Immunohistochemistry RNA isolation and qRT-PCR Western blot	ANXA2 mRNA and protein was significantly upregulated in GAC tissues ANXA2 expression correlated with TNM stage, lymph node and distant metastasis of GAC The expression of ANXA2 and E-cadherin were negatively correlated, hinting to an underlying mechanism by which ANXA2 contributes to the metastasis in GAC ANXA2 may represent a potential target for the treatment of GAC
[68]	Human GAC tissues Normal gastric epithelium adjacent to the tumors GC cell lines (ACP02 and ACP03)	Pathological analysis Immunohistochemistry Immunofluorescence RNA isolation and qRT-PCR Western blot	ANXA2 was up-regulated and GAL3 was reduced in GAC ANXA2 and GAL3 deregulated expression was associated with an invasive GC phenotype that may contribute to metastasis in GC patients
[67]	Human GC tissues Normal gastric epithelium adjacent to the tumors GC cell lines (HGC-27)	Pathological analysis Immunohistochemistry Transfection RNA isolation and qRT-PCR Western blot Cell Invasion, Migration and Gelatin Zymography Assays	ANXA2 was highly expressed in GC compared to normal controls and correlated to tumor size, histological differentiation, TNM stage, lymph node metastasis Upon inhibition of ANXA2, the migration and secretion of matrix metalloproteinases of HGC-27 cells were significantly decreased
[70]	GC tissues from The Cancer Genome Atlas (TCGA) database GC tissue microarray (HStmAde180Sur-06) GC cell lines (SGC-7901, MKN45, BGC-823) AGS cells	Pathological analysis Immunohistochemistry RNA Isolation and qRT-PCR Western blot Construction of Lentivirus and Infection Cell Proliferation, Colony Formation, Apoptosis, Matrigel Invasion and Transwell Migration Assays Wound Scratch Test Cell Cycle Analysis Bioinformatics Analysis	ANXA2 protein and mRNA expression was upregulated in GC tissues compared to adjacent tissues ANXA2 promoted GC cell migration and proliferation ANXA2 silencing in AGS cells inhibited their proliferation, invasion, migration, promoted apoptosis, and arrested the cell cycle Targeting ANXA2 may be a potential therapeutic strategy for GC patients
[71]	AGS cells MGC-803 cells Human GC cell lines (GES-1, HGC-27, SGC-7901, BGC-823, MKN-45, and MKN-74)	Pathological analysis Immunohistochemistry Immunofluorescence Western blot In vitro kinase and Proximity ligation Assays NOD-SCID mice	ANXA2 promotes GC progression and metastasis EphA2–YES1–ANXA2 axis is a novel pathway found to drive GC invasion and metastasis Targeting the EphA2–YES1–ANXA2 pathway could be an efficient treatment of GC

**Table 4 cancers-16-03764-t004:** The role of ANXA2 in colorectal tumors.

Ref.	Samples, Controls	Materials, Methods	Results
[73]	Human CRC cell lines (HT29-P, HT29-LM, KM12-C and KM12-SM)	Pathological analysis Immunohistochemistry Western blot	Overexpression of ANXA2 significantly correlated with CRC tumor size, histologic type, depth of invasion, and pTNM stage Tenascin-C overexpression correlated with CRC histologic type, depth of invasion, lymphatic invasion, venous invasion, lymph node metastasis and pTNM stage Expression of ANXA2 correlated with that of tenascin C ANXA2 and tenascin-C co-overexpression was an independent poor prognostic factor in CRC
[83]	Human CAC cell lines (BCS-TC2, BCS-TC2.2, HT-29) Human colon Caco-2 cells	Pathological analysis Immunohistochemistry ALP Activity Measurement Western blot FACS Analysis	Human CAC cell differentiation is associated with up-regulation and subcellular relocation of ANXA2 ANXA2 expression is highly dependent on the cell-growth state
[77]	Human CC cell line (HCT-116) Mouse colon cancer cell line (CA) Normal rat intestinal epithelial cell line (IEC-18) Mouse fibroblast cell line (Swiss 3T3)	Growth and Binding Assays qRT-PCR Peptide mapping Database mining Immunoprecipitation Western blot SELDI-TOF-MS	The membrane-associated ANXA2 binds PG/gastrins and partially mediates their growth effects
[74]	Human CC tissues Normal colorectal epithelium adjacent to the tumors	Pathological analysis Immunohistochemistry Tissue Microarrays Western blotIPG-2-DE and Image Analysis In-Gel Trypsin Digestion of Target Protein MALDI-TOF-MS Protein Identification and Database Analysis	CRC with lymph node metastases, compared to CRC without lymph node metastases, are characterized by increased expression of HSP-27, GST, and ANXA2, and decreased expression of L-FABP
[78]	Normal rat intestinal epithelial cell line (IEC-18)	Pathological analysis Immunohistochemistry Immunoprecipitation Western blot Transfection FACS Analysis Membrane Binding and Internalization assays	Progastrin binds to the cell surface-associated ANXA2 and is internalized via the clathrin-mediated endocytic pathway, resulting in activation of MAP Kinases Progastrin, covalently linked to sepharose beads, failed to activate p38MAPK/ERKs In the absence of clathrin and/or cell surface-associated ANXA2, progastrin fails to activate p38MAPK/ERKs Targeting clathrin-mediated endocytosis of progastrin may inhibit the carcinogenic effects of progastrin on intestinal cells
[85]	Human colon caco2 cell line (Annexin A2-/-caco2)	Annexin A2 knockout recombinant construction Transfection Scanning electron microscope (SEM) Transmission electron microscope (TEM) Immunofluorescence	ANXA2 enhances the invasion and metastasis of CC by increasing the expression and polymerization of F-actin and β-tubulin, and suppressing cell contact inhibition ANXA2 can be considered as an important target in the context of gene therapy of CC
[75]	Human CRC tissues Lymph Nodes with CRC metastases Normal colorectal epithelium adjacent to the tumors Human CRC cell lines (SW480, SW620, HT29 and HCT116) Human normal colonic epithelium cell line (NCM460)	Pathological analysis Immunohistochemistry Western blot	ANXA2 expression was significantly up-regulated in CRC cell lines and tissues ANXA2 expression correlated with poor differentiation, late stage, lymph node positivity, recurrence, and survival in patients with CRC ANXA2 expression and tumor location were independent factors in predicting overall survival ANXA2 expression was an independent factor predicting recurrence
[89]	Serum of patients with CC Serum from age and sex matched healthy controls	ELISA	Serum levels of ANXA2 in CC patients were significantly decreased compared to healthy controls, and decreased even further with increasing tumor size, TNM stage, depth of invasion, and lymph node and distant metastasis Potential use of serum levels of ANXA2 as a diagnostic biomarker in patients with CC
[84]	Human CRC tissues from patients with T3/4NxM0 CRC Human CRC cell lines (HCT-15, DLD-1, SW480, SW620, HCT-116 and LS174T)	Pathological analysis Immunohistochemistry Immunofluorescence Western blot Two-dimensional DIGE Cell invasion assay	Tumor deposits in pericolic fat, extramural vascular invasion, and the number of cells with ANXA2 membrane pattern in the tumor invasive edge had a significant influence on lymph node metastasis. The latter is a risk factor in stage II CC ANXA2 could be used as a biomarker of high-risk patients who would benefit from adjuvant therapy
[79]	Human CRC tissues Human CRA cell line (HT-29)	Pathological analysis Immunohistochemistry Immunofluorescence Western blot Cell Invasion, Proliferation, Migration and Wound Healing Assays RNA silencing Gene expression Analysis	ANXA2 is overexpressed in stage IV colorectal tumors and metastases ANXA2 and E-cadherin co-localize at cellular junctions and are internalized together upon TGF-ß treatment Existence of correlation between ANXA2 expression and specific cytoskeleton and motility-associated microstructures (CMS) of CRC (inverse relation between ANXA2 and CDH1) ANXA2 overexpression plays a pivotal role in CRC invasiveness through the activation of the Src/ANXA2/STAT3 axis
[86]	Human CRC cell line (caco2) Human HCC cell line (SMMC7721)	Pathological analysis Immunohistochemistry Immunofluorescence siRNA Designation and Transfection Semi-Quantitative RT-PCR Western blot Scanning Electron Microscope (SEM) Wound Healing, Transwell Chamber, MTT and Apoptosis Assays	Elevated ANXA2 expression enhances the malignancy of caco2 and SMMC7721 cells by remodeling theCMS, and by maintaining high cell proliferation
[81]	Human CC cell lines (HT-29, HCT116, SW480, LOVO) Human normal colonic epithelium cell line (NCM-460)	Pathological Analysis Bioinformatics Analysis Cell Grouping and Transfection Dual-Luciferase Reporter, RNA-FISH, Clone Formation, Cell Viability and Transwell Assays RIP qRT-PCR RNA Pull-Down Western blot Tumor Xenograft in BALB/c nude mice	LINC00460 silencing suppresses the proliferation, invasion, and tumorigenesis in CC cells by down-regulating ANXA2 via miR-433-3p LINC00460 shows high expression levels in CC cells and functions as a competing endogenous RNA (ceRNA) to sponge miR-433-3p to up-regulate ANXA2 The LINC00460/miR-433-3p/ANXA2 Axis regulates EMT in CC
[87]	Human CAC cell line (ANXA2-/-caco2)	Pathological analysis Immunohistochemistry Immunofluorescence qRT-PCR Western blot MTT, Colony formation, Wound healing, Transwell and Dual luciferase reporter Assays Tumor Xenograft in BALB/c nude mice	The 4 N-terminal sites of ANXA2 and, especially, the Y23 site, play important roles in maintaining ANXA2 phosphorylation and oncogenic effects in CAC The motility-associated microstructures of caco2 cells can be remodeled and their malignant behavior differentially inhibited by the mutation of the ANXA2 N-terminal sites Tumor growth can be repressed by the mutation of the ANXA2 N-terminal sites in vivo CAC malignancy can be suppressed by miR-206 targeting of ANXA2 in vitro and in vivo. The latter can be tested as a CAC therapy option
[88]	Human CC tissues Normal colorectal epithelium adjacent to the tumors Serum of CRC patients Serum from healthy controls	Pathological analysis Immunohistochemistry ELISA Chemiluminescence immunoassay for CA19.9 and CEA	Expression of ANXA2 is slightly higher in CRC tissues compared to normal adjacent paired tissues Levels of serum ANXA2 were significantly lower in CRC patients than in healthy controls, while the opposite was true for CA19.9 and CEA Serum ANXA2 or, joint analysis of serum ANXA2 with CEA, could be used as auxiliary diagnostic markers in CRC
[80]	Human CRC cell lines (HCT116, SNU-C1, LoVo and SW480) Human normal colonic epithelium cell line (NCM460)	Western blot Transfection RT-qPCR Cell Invasion, Proliferation (EdU), Counting (CCK-8) and Wound Healing Assays	TAGLN2 overexpression promotes CRC cell proliferation, migration and invasion TAGLN2 activates ANXA2 and STAT3, facilitating EMT
[76]	Human CC tissues Normal colon epithelium Human colon adenoma tissues	Pathological analysis Immunohistochemistry	ANXA2 expression was significantly increased in CRC compared to normal mucosa and colonic adenomas and was associated with aggressive cancer features High ANXA2 expression correlated with high-grade dysplasia and the tubulovillous adenoma type
[82]	Human colon polyp tissues Normal adjacent colon epithelium	Gene expression analysis Cdna Synthesis qRT-PCR MicroRNA-Interaction-Targets (MITs) Network Analysis Enrichment Analysis	ANXA2 and LINC00460 genes were significantly overexpressed in patients with colorectal polyps, especially in high-risk ones, and were associated with poor clinical outcomes ANXA2 and LINC00460 was deregulated in polyps compared to normal tissues

**Table 5 cancers-16-03764-t005:** The role of ANXA2 in liver tumors.

Ref.	Samples, Controls	Materials, Methods	Results
[90]	Human HCC tissues Cirrhotic liver tissues adjacent to the tumors	Pathological analysis Immunohistochemistry Immunofluorescence Northern blot Western blot	Overexpression of ANXA2 in the sinusoidal endothelium and in the malignant hepatocytes of HCC Possible role of ANXA2 in angiogenesis in HCC
[99]	Human HCC tissues Normal liver tissues without viral infection	Pathological analysis Immunohistochemistry RNA extraction qRT-PCR Gene network analysis	Integrin and Akt/NF-κB signaling pathways and osteopontin (SPP1), glypican-3, ANXA2, S100A10 and vimentin play pivotal roles in HCC Co-localization of ANXA2 and S100A10 was found in human HCC tissues
[91]	Human HCC tissues Liver tissues from patients with chronic hepatitis	Pathological analysis Immunohistochemistry SDS-PAGE RNA Extraction Northern blot Western blot Partial purification Immunoprecipitation Densitometry	Overexpression of ANXA2 at both transcriptional and translational levels in tumorous and non-tumorous regions of HCC, often with greater expression in the former ANXA2 is almost undetectable in either normal or chronic hepatitis liver tissues The expression of ANXA2 was mainly localized in cancer cells, especially in poorly differentiated HCC ANXA2 was tyrosine-phosphorylated in HCC
[96]	Human HCC tissues Normal liver tissues Serum of HCC patients Serum from healthy controls	RT-PCR Western blot ELISA	Expression of ANXA2 was significantly elevated in the liver tissues and serum of HCC patients when compared with those of normal individuals
[103]	Human CCA cell lines (KKU-M156, KKU-100, KKU-139 and KKU-M213) linked with Opisthorchis viverrini infection Human nonmalignant cholangiocyte cell line (H69) Human intrahepatic CCA tissues linked with Opisthorchis viverrini infection	Pathological analysis Immunohistochemistry 2-DE and image analysis MALDI-TOF MS Western blot Tissue microarray	ANXA2 is expressed both in the cytoplasm and cell membrane of hyperplastic epithelia and CCA but not in normal cholangiocytes and hepatocytes Increased expression of ANXA2 did not correlate with the CCA histological subtype or tumor size High expression of ANXA2 was associated with metastasis and lymphatic and perineural invasion, as well as poor prognosis and shorter survival time Up-regulated ANXA2 may serve as a prognostic marker for invasion, metastasis and survival in O.viverrini-associated CCA
[92]	Human hepatocellular nodules, including macroregenerative nodules (MRN), dysplastic nodules (DN), HCC, hepatocellular adenomas (HCA) and focal nodular hyperplasia (FNH)	Pathological Analysis Immunohistochemistry Tissue microarray	Adding ANXA2 to the markers panel for the detection of well-differentiated HCC (GPC3, GS, and HSP70) increases diagnostic reliability in liver biopsies The combinations ANXA2-GPC3 and ANXA2-GS showed higher diagnostic value compared to the other marker combinations
[100]	Human HCC cell lines (SMMC-7721, 7721-si18G, T7721, Huh-7, and MHCC97-H)	Immunofluorescence RNA Interference Western blot Northern blot	The interaction of HAb18G/CD147 with ANXA2 is involved in the mesenchymal and amoeboid movement of HCC cells, influencing their metastasis potential and invasion ability HAb18G/CD147 inhibits the Rho/ROCK signaling pathway and the amoeboid movement of HCC cells by inhibiting the phosphorylation of ANXA2 HAb18G/CD147 promotes membrane localization of WAVE2 and activation of Rac1 in HCC cells through the Integrin-FAK-PI3K/PIP3 signaling pathway
[97]	Human HCC cell lines (HepG2, Hep3B, SK-HEP-1, Bel-7404, SMMC7721 and HLE) Normal human liver cell line (HL-7702) Human HCC tissues Normal human liver tissues adjacent to the tumors Serum from HCC patients Serum from patients with hepatitis, post-hepatitic cirrhosis, benign liver tumors Serum from healthy controls	Pathological Analysis Immunohistochemistry Immunofluorescence 2-DE separation and protein identification ELISA Pepscan analysis Western blot p21-HBx transgenic mice	ANXA2 was up-regulated in both tissues and sera of HCC patients in AFP positive and negative cases Combining ANXA2 and AFP improved the diagnosis of early HCC ANXA2 expression was substantially elevated in mice with HCC ANXA2 may be an independent and discriminative serological candidate biomarker of HBV-related HCC, especially in early-stage cases with normal serum AFP
[93]	Human HCC cell lines (HepG2, SMMC-7721, SMMC-7402, and MHCC97-H) Normal human hepatocyte cell line (LO2)	Pathological Analysis Immunohistochemistry Transfection RNA isolation and cDNA synthesis Immunofluorescence Western blot qRT-PCR Cell cycle, Cell proliferation, Transwell, in vitro wound- healing and Xenograft tumor-growth assay BALB/C nude mice	ANXA2 is up-regulated in HCC cells with high metastatic potential and invasion ability shRNA-mediated silencing of ANXA2 suppresses the invasion, migration, and tumorigenic potential of HCC cells
[94]	Human cancerous, paracancerous and noncancerous tissues from patients after surgery for HCC	Pathological analysis Immunohistochemistry Immunofluorescence Transfection RNA isolation qPCR Western blot Cell cycle, Cell proliferation, Transwell and Xenograft tumor-growth assay BALB/C nude mice	ANXA2 expression correlated with HBV, extrahepatic metastasis, portal vein thrombosis and high metastatic potential Inhibition of ANXA2 arrested the cell cycle in vitro and inhibited tumor growth in vivo
[95]	Human HCC tissues Human liver tissues from patients with benign liver disease (BLD) Normal human liver tissues adjacent to the tumors Serum from HCC patients Serum from patients with BLD	Pathological analysis Immunohistochemistry Enzyme-linked immunosorbent assay RNA isolation qRT-PCR Tissue microarrays	ANXA2 expression was frequently up-regulated in the serum and in the tissues of HCC patients The increasing ANXA2 expression correlated with intrahepatic metastasis, portal vein thrombosis, and higher TNM stages ANXA2 acted as a new adverse independent prognostic factor for HCC
[101]	Human HCC tissues Human HCC cell line (SMMC-7721) HepG2, A549 and HCT-8 cell lines CD147-knockout SMMC-7721 cell line (K-7721)	Pathological analysis Immunohistochemistry Immunofluorescence Western blot qRT-PCR Nuclear/ cytoplasmic fractionation FRET, In vitro kinase, in vitro wound-healing and in vivo metastasis Assays Surface plasmon resonance (SPR) Fluorometric analysis Live-cell imaging	CD147 physically interacted with the N-terminal domain of ANXA2 and decreased its phosphorylation on tyrosine 23 p-Annexin A2 promoted the expression of DOCK3, a negative regulator of WAVE2 expression by inhibiting β-catenin signaling CD147 promotes tumor cell movement and metastasis via direct interaction with the ANXA2 and DOCK3-β-catenin-WAVE2 signaling axis
[102]	Human HCC tissues Human HCC cell lines (HepG2, Hep3B, Huh 7, MHCC97H, PLC/PRF/5 and HCCLM3)	Pathological analysis Immunohistochemistry Western blot RT-PCR Tissue microarrays Lentivirus production and transduction of target cells Co-immunoprecipitation 2D-LC-MS/MS In vivo tumor growth, Ubiquitination, Cell proliferation and invasion Assays	High level of ANXA2 expression correlates with poor prognosis of HCC patients UBAP2 is significantly down-regulated in HCC tissues compared with adjacent normal tissues Overexpression of UBAP2 promotes ubiquitination and degradation of ANXA2 and thus impairs the progression of HCC Enforced ANXA2 expression negates the inhibited invasion induced by the overexpression of UBAP2 UBAP2 may be a novel prognostic biomarker and a therapeutic target for HCC
[98]	Serum from patients with early-stage HCC and chronic liver disease Serum from patients with chronic liver disease without HCC Serum from healthy, age- and sex-matched controls	ELISA	Serum ANXA2 levels were significantly elevated in patients with HCC compared to those with chronic liver disease, as well as the healthy controls Serum ANXA2 may serve as a biomarker for the early detection of HCC, since it shows higher sensitivity, specificity, and positive and negative predictive value than AFP

**Table 6 cancers-16-03764-t006:** The role of ANXA2 in pancreatic tumors.

Ref.	Samples, Controls	Materials, Methods	Results
[104]	Tissue from primary and metastatic pancreatic tumors Normal human pancreatic tissues Human PDAC cell lines (HPAF, CD11 and CD18, Panc-1 and Panc-89)	Pathological analysis Immunohistochemistry Western blot	Higher levels of ANXA2 were found in pancreatic tumors compared to normal pancreas
[109]	Human PC cell lines (SK-PC-1 and PANC-1)	Pathological analysis Immunohistochemistry Western blot Immunoprecipitation	t-PA specifically binds to ANXA2 on the extracellular membrane of PC cells, where it activates local plasmin production and tumor cell invasion
[111]	Human PC cell lines (ASPC-1, BxPC-3, Capan-1, Colo-357, MiaPaCa2, Panc-1 and T3M4) Human PDAC tissues Normal human pancreatic tissues	Pathological analysis Immunohistochemistry Western blot ELISA qRT-PCR DNA array Analysis of TNC splicing variants	The expression of Tenascin C and cell surface ANXA2 increases in the progression from low-grade PanIN to PC Tenascin C is expressed in preinvasive pancreatic lesions and diffusely in the stroma of PC, while ANXA2 shows a preferential membranous localization on the tumor cells Pancreatic stellate cells are a source of Tenascin C in pancreatic tissues, possibly under the influence of soluble factors released by the tumor cells
[105]	Human PDAC tissues Normal human pancreatic tissues	Pathological analysis Immunohistochemistry Western blot 2DE and image analysis In-gel digestion and mass spectrometric analysis	Nucleotide diphosphatase kinase (NDPK) and ANXA2 were overexpressed in PDAC, and may be used as molecular biomarkers for diagnosis or as therapeutic targets
[112]	Human PDAC tissues Normal human pancreatic tissues Human PC cell lines (WT-MIA PaCa-2) Gemcitabine-resistant human PC cell lines (GEM-MIA PaCa-2)	Pathological analysis Immunohistochemistry Protein Extraction Western blot qRT-PCR Gene Knockdown by siRNA Cell Viability and Cytotoxicity Assay	ANXA2 was up-regulated in the gemcitabine-resistant PC cell line ANXA2 overexpression in cancer cells was associated with rapid recurrence after gemcitabine adjuvant chemotherapy in postoperative patients ANXA2 IHC degree of staining correlated with the disease-free survival of patients who underwent adjuvant gemcitabine treatment after curative surgery Inhibition of ANXA2 expression in gemcitabine-resistant PC cells increases the cytotoxic efficacy of gemcitabine
[110]	Human PC cell lines (Panc-1, MiaPaCa-2 and Suit-2)	Pathological analysis Immunohistochemistry Immunofluorescence Western blot Gene Knockdown by siRNA Immunoprecipitation 2DE protein identification SDS–PAGE	A positive relationship was found between high levels of cytoplasmic S100A6 and the localization of ANXA2 to the cell membrane that promotes PC cell motility
[114,115]	Human PDAC tissues Serum from patients with PDAC receiving an allogeneic GM-CSF secreting tumor vaccine (GVAX) Human PDAC cell lines Human fibroblast cell line Mouse Panc02 cells Female C57Bl/6 mice	Pathological analysis Immunohistochemistry Immunofluorescence ELISA RT-PCR Immunoprecipitation and immunoblot analysis Whole cell extract and cell fractionation DNA cloning and plasmid constructions Plasmid transfection, lentiviral infection and RNA interference Cell invasion assays	Cell surface ANXA2 increases with PDAC progression Phosphorylation of ANXA2 at Tyr23 promotes its localization in the cell surface which is required for the TGFβ-induced, Rho-mediated EMT of PDAC cells that increases their invasion capacity Expression and tyrosine phosphorylation of ANXA2 are required for PDAC metastases in vivo Inhibition of ANXA2 suppresses in vitro invasion and in vivo metastases of PDAC, prolonging survival in a mouse model of PDAC Post-vaccination patient sera inhibit in vitro invasion of PDAC cells Identification of ANXA2 as candidate PDAC biomarker and antigenic target for the development of therapeutic antibodies and T-cell immunotherapy
[106]	Human PDAC tissues Normal human pancreatic tissues	Pathological analysis Immunohistochemistry	ANXA2 showed higher expression in PDAC tissues compared with adjacent normal tissues and was associated with the pathological grade of PDAC, indicating that Annexin A2 may be used as a diagnostic biomarker PDAC tissues had higher microvessel density than that of adjacent normal tissues and ANXA2 positively correlated with this Microvessel density was associated with the tumor grade and TNM stage and was an independent risk factor for PDAC patients
[107]	Human PDAC cell lines (PANC-1, PANC10.05, BxPC-3, MiaPaca-2, CFPAC-1, Colo357 and Capan-1) HEK293FT cells	Pathological analysis Immunohistochemistry Immunofluorescence mRNA isolation and qRT-PCR Luciferase, Plasmin production and Tube formation Assays SCID mice	KRAS and ANXA2 are up-regulated in PDAC specimens, and their expression inversely correlates with miR-206 expression miR-206 regulates different hallmarks of cancer by targeting multiple oncogenic routes, involving KRAS-induced NF-κB signaling, ANXA2 and VEGFC
[113]	Human PC cell lines (HeLa, HEK-293, SW480 and MiaPaca2 cells)	Pathological analysis Immunohistochemistry qRT-PCR Immunoprecipitation His-tagged and GST- tagged protein pull-down siRNA and transduction Subcellular fractionation Quantification of IL-6 Luciferase reporter, Cell viability and LDH cytotoxicity Assays	ANXA2 interacts with the p50 subunit of NF-κB, increasing its transcriptional activity and up-regulating anti-apoptotic genes, including that encoding IL-6 ANXA2 knockdown affects the transcriptional activity of NF-κB and the viability of PDAC cell Overexpression of ANXA2 may be involved in gemcitabine resistance in PC cells ANXA2 is a potential biomarker of drug resistance and a candidate therapeutic target for the treatment of PC
[116]	Human PDAC tissues from patients who underwent pancreaticoduodenectomy and received adjuvant chemoradiation therapy PDAC Tissue microarray from specimens of a JHMI IRB–approved rapid autopsy protocol KPC and KPCA−/−cell lines	Pathological analysis Immunohistochemistry Immunofluorescence qRT-PCR Western blot Plasmid transfection and RNA interference Tissue Microarrays Sema3D ELISA Co-immunoprecipitation Invasion, AP binding and TUNEL Assays KPC mice	ANXA2 is essential for PDAC metastasis in a transgenic mouse model The gene expression of the semaphorin 3D is increased in PDAC ANXA2 promoted the secretion of semaphorin 3D from PDAC cells, which co-immunoprecipitated with PlxnD1 on PDAC cells Semaphorin 3D and PlxnD1 increase the invasion capacity of PDAC Knockdown of semaphorin 3D decreases the invasion and metastatic capacity of PDAC cells and prolongs the survival of PDAC-bearing mice Antibodies against ANXA2 in the sera of some patients after surgical resection of PDAC are associated with longer recurrence-free survival Possible value of developing new therapies targeting ANXA2 and semaphorin 3D as an adjuvant treatment for PDAC after local resection
[117]	Human PDAC tissues KPC and KPCA−/−cell lines	Pathological analysis Immunohistochemistry KPC mice C57BL/6J (B6) mice AnxA2 KO mice	High stromal ANXA2 expression was associated with significantly reduced disease-free and overall survival ANXA2 expression may serve as a prognostic biomarker of survival in patients with PDAC B6 mice injected with KPC cells demonstrated decreased median survival compared to those injected with KPCA cells, whereas there was no survival difference in the ANXA2 KO mice
[108]	Human PDAC tissues Human PDAC cell line (PANC-1) Murine PanIN cells (KC), PDAC cells (KPC1 and KPC2), and liver metastatic PDAC cells (KPCLiv)	Pathological analysis Immunohistochemistry siRNA Western blot rTNC treatment 3D organotypic cell culture Pancreatosphere formation, Cell invasion and anoikis Assays	ANXA2 is expressed at high levels in murine primary PC cells The ANXA2- Tenascin C interaction sequentially regulated EMT, invasion and metastatic colonization in PDAC ANXA2 and Tenascin C maintain the stem-like characteristics of PDAC cells The ANXA2–Tenascin C interaction contributes to anoikic resistance in PDAC cells High expression of ANXA2 and Tenascin C positively correlates with distant metastasis, hematogenous and peritoneal recurrence and poor prognosis in PDAC

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
