# Peer review of "Annexin A2 in Tumors of the Gastrointestinal Tract, Liver, and Pancreas"

_cancers, 2024, doi:10.3390/cancers16223764_

Round 1

Reviewer 1 Report

Comments and Suggestions for Authors

I recommend the publication of manuscript cancers-3240699 “The Role of Annexin A2 in Tumors of the Digestive System” in Cancers journal after a major revision. In my opinion, English language is understandable, and the work does not require language editing. Authors review very interesting topic about the impact of Annexin A2 on inflammation and pathophysiology in some digestive system cancers, such as esophagus, stomach, colorectum, liver or pancreas. Below I present the most important remarks regarding the work.

MAJOR ISSUES:

1. The title does not reflect what is described in the text. The human digestive system consists of the gastrointestinal tract plus the accessory organs of digestion (e.g. the tongue, salivary glands, etc.). Please change the title to one that better fits the elements described in the manuscript, I suggest specifying the title, e.g “The Role of Annexin A2 in Selected Digestive System Tumors” or other, more specific one.

2. Although the work is extensive and contains a large amount of references, it seems chaotic.  Text formatting needs improvement. In the Introduction, please use subchapters describing 1.1. Structure and Cellular Localization of Annexin A2, 1.2. Regulation of Annexin A2, 1.3. Physiological Functions and Pathological Implications of Annexin A2, or please create an additional chapter after Introduction “2. The protein Annexin A2 and its functions” and describe these subchapters there.

3. Same in chapter 3.The role of Annexin A2 in DS tumor, apply subsections 3.1. Annexin A2 in esophagus tumor, 3.2. Annexin A2 in stomach tumor, etc.

4. Clearly describe the inclusion and exclusion criteria.

5. Please add more information on whether there is research on ANXA2 as a treatment target in specific cancer types: esophagus, stomach, colorectum, liver or pancreas (by blocking signals etc.)?

MINOR ISSUES:

1. Line 54: authors wrote “as well as in various tumors”. List examples here and cite relevant literature.

2. In “Pathological Implications” examples should not be listed in the form of sub-items, use continuous description in the text or I suggest creating a chart/ figure here.

3. Changes in tables: a) use the table format provided in the Cancers journal template; b) remove the "Signaling pathways" column; c) replace the reference column with [59] instead of Qi YJ et al. [59] - there will be more space for the remaining large columns; d) consider designing a horizontal column, or adjust the font size so that the tables are compact and more readable;

Author Response

We would like to thank the reviewer for their kind and helpful comments. Please find a point-by-point response to your comments below:

- MAJOR ISSUES:

  1. The title does not reflect what is described in the text. The human digestive system consists of the gastrointestinal tract plus the accessory organs of digestion (e.g. the tongue, salivary glands, etc.). Please change the title to one that better fits the elements described in the manuscript, I suggest specifying the title, e.g “The Role of Annexin A2 in Selected Digestive System Tumors” or other, more specific one.

Thank you for your comment. We have modified the title into “ANXA2 in Tumors of the Gastrointestinal Tract, Liver and Pancreas” to better reflect what is described in the text.

  1. Although the work is extensive and contains a large amount of references, it seems chaotic.  Text formatting needs improvement. In the Introduction, please use subchapters describing 1.1. Structure and Cellular Localization of Annexin A2, 1.2. Regulation of Annexin A2, 1.3. Physiological Functions and Pathological Implications of Annexin A2, or please create an additional chapter after Introduction “2. The protein Annexin A2 and its functions” and describe these subchapters there.

Thank you for helping us make our work easier to follow. We have now divided our introduction into subchapters as suggested.

  1. Same in chapter 3.The role of Annexin A2 in DS tumor, apply subsections 3.1. Annexin A2 in esophagustumor, 3.2. Annexin A2 in stomach tumor, etc.

Same as introduction, we have now divided chapter 3 into subsections as suggested.

  1. Clearly describe the inclusion and exclusion criteria.

We have modified the second paragraph of chapter 2 (methods) to clearly describe our study’s inclusion and exclusion criteria as follows:«Specifically, original studies and reviews in English language concerning the biological functions of ANXA2 in GI tract, liver and pancreas in humans were eligible for inclusion. Letters, comments, editorials and case reports, as well as abstracts or conference abstracts without full text available, studies written in languages other than English were excluded from this review »

  1. Please add more information on whether there is research on ANXA2 as a treatment target in specific cancer types: esophagus, stomach, colorectum, liver or pancreas (by blocking signals etc.)?

Thank you for your comments. For each organ studied, we have now added a paragraph at the end of each corresponding section that includes all references that highlight the importance of ANXA2 as a diagnostic and prognostic biomarker, as well as its role as a therapeutic tool. Additionally, we have created an additional chapter (4. Therapeutic implications of Annexin A2), summarizing all the potential therapeutic applications of ANXA2.

- MINOR ISSUES:

  1. Line 54: authors wrote “as well as in various tumors”. List examples here and cite relevant literature.

The phrase has been omitted since this section concerns ANXA2 in normal tissues.

  1. In “Pathological Implications” examples should not be listed in the form of sub-items, use continuous description in the text or I suggest creating a chart/ figure here.

Thank you for the suggestion. In the revised version, we have added a table entitled “The pro-oncogenic role of ANXA2” covering the subject (Table 1).

  1. Changes in tables: a) use the table format provided in the Cancers journal template; b) remove the "Signaling pathways" column; c) replace the reference column with [59] instead of Qi YJ et al. [59] - there will be more space for the remaining large columns; d) consider designing a horizontal column, or adjust the font size so that the tables are compact and more readable;

The revised manuscript’s tables have been amended according to all suggested corrections.

Reviewer 2 Report

Comments and Suggestions for Authors

1. This review is expected to conduct a concise analysis of the expression and function of ANXA2 in various types of digestive system tumors, instead of simply aggregating them.
2. For each cited literature, more emphasis should be placed on its significance rather than enumerating all the results in detail.
3. Is it necessary to repeat similar contents in each type of tumor, or is it better to emphasize the specificity of the expression and function of ANXA2?
4. The table presenting the literature situation can be placed in the supplementary data.
5. The analysis of the correlation with clinical applications is still inadequate.

Comments on the Quality of English Language

It is better to ask a professional English Editor to review the manuscript.

Author Response

We would like to sincerely thank the reviewer for their kind and helpful comments that enabled us to further improve our manuscript. Please find a response to your comments below:

  1. This review is expected to conduct a concise analysis of the expression and function of ANXA2 in various types of digestive system tumors, instead of simply aggregating them.

Thank you for your comments. For each organ studied, we have now added a paragraph at the end of each corresponding section that includes all references that highlight the importance of ANXA2 as a diagnostic and prognostic biomarker, as well as its role as a therapeutic tool.

  1. For each cited literature, more emphasis should be placed on its significance rather than enumerating all the results in detail.

Thank you for stressing out this important point. To highlight the significance of ANXA2, we emphasized more on its potential clinical applications, as analyzed in comments 1 and 5.

  1. Is it necessary to repeat similar contents in each type of tumor, or is it better to emphasize the specificity of the expression and function of ANXA2?

This is a valid point. Nevertheless, our aim in writing this systematic review was to present to our readers the important data of all the research articles included in our review. In that way we hope we will aid them by removing the need for them to extract the data themselves, a process that can be cumbersome.

  1. The table presenting the literature situation can be placed in the supplementary data.

We understand that the tables are extensive and can seem chaotic. To solve that we removed some of its elements and made its overall format clearer. Thank you for helping us make our work easier to follow.

  1. The analysis of the correlation with clinical applications is still inadequate.

In addition to the concluding paragraph for each organ, we have created an additional chapter (4. Therapeutic implications of Annexin A2), summarizing all the potential therapeutic applications of ANXA2.

Reviewer 3 Report

Comments and Suggestions for Authors

Dear  Authors,

Thank you for your submission. This is a very interesting scientific paper and well comprehensive. Please add one graphical abstract for the benefit of the paper and of course for the readers. 

Author Response

We would like to sincerely thank the reviewer for their kind and helpful comment that enabled us to further improve our manuscript. Please find a response to your comment below:

Dear  Authors,

Thank you for your submission. This is a very interesting scientific paper and well comprehensive. Please add one graphical abstract for the benefit of the paper and of course for the readers. 

Reply: Thank you for your comments. As per your instructions, a graphical abstract has been added in our manuscript, directly after the abstract.

Round 2

Reviewer 1 Report

Comments and Suggestions for Authors

Please correct the format of the tables, in this version they are unreadable.

The authors should improve the formatting of tables, preferably increase their width (maximum to the size of the page), eliminate the bullet (decorative bullet point) from tables, which will also increase the space in the column, reduce the font size a bit and enter only essential sentences in the columns - as possible.Reading a table that is 3-4 pages long is distracting and it doesn't look good. 

Author Response

Dear Reviewer 1,

Thank you for your valuable comments. As instructed, we have increased the width of the tables, we have eliminated all the bullets and have reduced the font size. We hope that this way the tables will be easier to read.

We remain available for anything further required.

Kind regards,

Dr. Emmanouil I. Kapetanakis

Thoracic Surgeon